# Lossy P-LDPC Codes for Compressing General Sources Using Neural Networks

**DOI:** 10.3390/e25020252

**Published:** 2023-01-30

**Authors:** Jinkai Ren, Dan Song, Huihui Wu, Lin Wang

**Affiliations:** 1Department of Information and Communication Engineering, Xiamen University, Xiamen 361005, China; 2Department of Electrical and Computer Engineering, McGill University, Montreal, QC H4H 1R3, Canada

**Keywords:** lossy compression, neural networks, general sources, P-LDPC codes, distortion-rate performance

## Abstract

It is challenging to design an efficient lossy compression scheme for complicated sources based on block codes, especially to approach the theoretical distortion-rate limit. In this paper, a lossy compression scheme is proposed for Gaussian and Laplacian sources. In this scheme, a new route using “transformation-quantization” was designed to replace the conventional “quantization-compression”. The proposed scheme utilizes neural networks for transformation and lossy protograph low-density parity-check codes for quantization. To ensure the system’s feasibility, some problems existing in the neural networks were resolved, including parameter updating and the propagation optimization. Simulation results demonstrated good distortion-rate performance.

## 1. Introduction

It is known that low-density parity-check (LDPC) codes have capacity-approaching performance as channel codes [1,2]. As a consequence, LDPC codes have been widely used in modern communication standards and in industrial applications. To simplify the structure, more constructive protograph LDPC (P-LDPC) codes are introduced with lower decoding complexity [3]. Furthermore, P-LDPC codes have good coding properties, and they can be easily optimized by convergence analysis of mutual information [4].

The duality between the lossy source coding and channel decoding is found with the compression of the Bernoulli sources [5,6]. Existing works show that LDPC codes have been developed for compressing the binary symmetric sources [7,8]. For instance, the belief propagation (BP) and its modifications are employed to be good candidates for lossy source coding [9,10].

Following this fact, more constructive P-LDPC code was introduced to replace the LDPC code. In [11], the BP algorithm based on the P-LDPC code was firstly proposed to compress the binary source with good performance. Then, Ref. [12] demonstrated the P-LDPC-based encoding algorithm can simultaneously overcome the source-compression distortion and channel-noise impact. Furthermore, the BP based on the P-LDPC code was firstly used for Gaussian source compression in [13]. In these cases, one P-LDPC code could be used simultaneously in source coding and channel coding to implement different functions. This is friendly to hardware manufacturing by reusing the P-LDPC decoding chip.

However, the aforementioned conventional algorithms and methods are complicated and time-consuming. It should be noted that the BP algorithm needs more iterations in the coding procedure. Moreover, the “quantization-compression” scheme is complicated, including two steps; see [13]. First, the Gaussian source is quantized to a binary sequence by using a high rate quantizer. Then, the binary sequence is compressed by P-LDPC codes.

It should be noted that the lossy compression inevitably brings bit errors, and each quantized bit contains different information. This uneven distribution of bit information results in large distortion in the source reconstruction. One solution is to use the multilevel coding (MLC) with set partitioning and rate allocation at different levels to homogenize distortions [13]. However, the MLC still has some problems. First, the MLC is more complex than the single-level coding structure. Second, the set partitioning cannot completely homogenize distortions by increasing coding levels. Third, it is difficult to find an optimal rate-allocation scheme. Table 1 briefly describes the mentioned literature, and Table 2 compares the methods and sources of the literature.

To conquer the aforementioned problems, a new route of “transformation-quantization” is proposed to design an efficient lossy compression based on P-LDPC codes. With the rapid development of deep learning, neural networks are used in a variety of tasks due to their excellent ability for information extraction, and they are used in data compression fields such as image and video compression [14,15,16]. The authors of [17] optimized autoencoders for lossy image compression, and the paper [18] presents a learned image compression system based on generative adversarial networks. Moreover, the paper [19] proposes an enhanced invertible encoding network with invertible neural networks to mitigate the information-loss problem for better compression. Recently, the diffusion model is also used in image compression fields [20]. In the aforementioned papers, the image data are modeled as the Gaussian source. To extend the source properties, the general source is considered to be compressed by the neural networks in this work. Here, the “transformation-quantization” scheme combines neural networks and lossy P-LDPC (NN-LP-LDPC) codes, as shown in Figure 1, where the transformation using neural networks and quantization using lossy P-LDPC codes modules are unified to be the encoder.

In Figure 1, the encoder includes the transformation and quantization, and there is feedback from quantization module to transformation module. The decoder contains the source reconstruction. Here, the neural networks are employed in the transformation and reconstruction modules, and the quantization is designed with a restrict minimum distortion (RMD) algorithm based on the P-LDPC code. In this encoder, the transformation performs a nonlinear conversion on the continuous sequences, and the quantization converts the continuous sequences into binary sequences. In the decoder, the binary sequences are reconstructed as continuous sequences.

Some key issues are resolved by the NN-LP-LDPC system. First, the conventional BP algorithm cannot be directly used for compressing continuous sources, since it will bring larger distortion. The RMD algorithm is proposed to improve the quantizer. Second, neither the BP nor the RMD has an index; thus, the source is restored without reference. The powerful function-fitting ability of the neural networks serves for the reconstruction to overcome this problem. Third, the quantization function is difficult to be implemented by the neural networks. Since the derivative function of the quantizer is almost zero, the gradient backpropagation is interrupted, and the coefficients of the neural networks cannot be updated. To address the problem, a new derivative function is evaluated to successfully realize the gradient backpropagation from the previous layer. Finally, the adaptation of the compression rate between the quantization and transformation modules is also important. A multi-level feedback mechanism is designed to provide the prior output as the input of the next quantizer. In this way, the sequence length increases with the growing number of quantizers. In addition, the compression rates can be changed by the mask in the RMD algorithm.

The main contributions are summarized as follows.

(1) The NN-LP-LDPC system is proposed for compressing general continuous sources, which complements the vacancy of continuous source compression based on binary LDPC codes. Furthermore, the proposed system is robust to different source distributions.

(2) A new route of “transform-quantization” is designed for the NN-LP-LDPC system, which efficiently replaces the conventional “quantization-compression” scheme. The simulation results validate the usefulness of new route. In addition, they provide a good reference to diversely process different kinds of sources.

(3) The P-LDPC code was efficiently combined with the neural networks, by which the emerging technical problems were successfully resolved. This enormously enriches the designing methodology for the source coding based on the P-LDPC code.

The rest of this paper is organized as follows: Section 2 introduces the proposed scheme. Some key techniques and system optimization are discussed in Section 3. The simulation results and analyses are shown in Section 4. Section 5 concludes this work.

## 2. NN-LP-LDPC System

As shown in Figure 1, a memoryless continuous sequence s of length *n* is input into the transformation module, and its output is $s^{\prime }$ of length mn, where *m* and *n* are integers. Then, the continuous sequence $s^{\prime }$ is quantized as a binary sequence q with the same length mn. It should be noted that the quantized q is also the feedback information to transformation module. Next, the updated q seves as the output of the encoder, and it is used to reconstruct the source $\hat{s}$.

### 2.1. Transformation Module

The encoder consists of the transformation and quantization modules that are detailed in Figure 2. First, the continuous sequence $s=\{s_{1},s_{2},\ldots ,s_{n}\}$ is input, and it is transformed as $s_{i}^{\prime }=\left\{s_{i,1}^{\prime },s_{i,2}^{\prime },\ldots ,s_{i,n}^{\prime }\right\}$, where i∈[1,m], and [a,b] represents the set of integer numbers from *a* to *b*. The sequence $s_{i}^{\prime }$ is sent to the *i*th quantizer $Q_{i}$. Then, the continuous sequence $s_{i}^{\prime }$ is quantized as a binary sequence $q_{i}=\left\{q_{i,1},q_{i,2},\ldots ,q_{i,n}\right\}$. For i∈[1,m−1], each $q_{i}$ is fed back to the MLP of the transformation module, and $q_{i}^{\prime }$ is the result. The consolidated s and $q_{i}^{\prime }$ are as the input of the CNN, and the output $s_{i+1}^{\prime }$ is as the input of i+1th quantizer $Q_{i+1}$.

The transformation module contains the multi-layer perceptron (MLP) and the convolutional neural network (CNN). An MLP provides a nonlinear transformation to change s into $s_{i}^{\prime }$, and the quantization function $Q_{i}$ obtains the corresponding $q_{i}$. After that, $q_{i}$ is returned to another MLP and transformed to $q_{i}^{\prime }$, and then it is appended on the s, which is presented as: (1)$$s\;⊕\;q{’}_{i}\Rightarrow \left[\begin{array}{c} s\\ q_{1}^{\prime }\\ \vdots \\ q_{i}^{\prime } \end{array}\right],$$
Then, the appended result increases one dimension of the channel, and it is sent to CNN as the input. The resulting ${s^{\prime }}_{i+1}$ is as the input of $Q_{i+1}$, and $q_{i+1}$ is acquired.

As shown in Figure 3, the MLP is the structure of the fully-connected (FC) layer, including an input layer, an output layer and a hidden layer. The FC layer is expressed as: (2)$$Y=XW_{h}+b_{h},$$
where $X\in R^{p\times n}$ is a small batch of inputs; *p* represents the batch size; the dimensions of the input are *n*; $Y\in R^{p\times k}$ is the output of dimension *k*; $W_{h}\in R^{n\times k}$ and $b_{h}\in R^{1\times k}$ are the weight and bias parameters, respectively; and R indicates the set of real numbers. It should be noted that X and Y refer to the input and output variables in general, respectively.

The activation functions are used to implement nonlinear transformations in the hidden layers. For the MLPs, the active function of the hidden layer is ReLU, which is shown as follows: (3)ReLU(x)=max(x,0)

Generally, the CNN contains several convolution layers. It is commonly used in the field of computer vision with a 2D stride and kernels [21,22], and the 1D convolution is confronted with the sequence data [23]. Furthermore, the convolution kernel larger than one is designed to increase the receptive field [24,25]. However, the memoryless continuous source has no spatial locality; hence, a larger field is unnecessary. In addition, it is convenient that the CNN in the transformation module processes multi-channel data, where the kernel size is one and the pooling layer is not needed.

Considering the aforementioned facts, the CNN only has a 1D convolutional layer with kernel size one, as shown in Figure 4. Similarly to the MLP, the CNN has one hidden layer and uses ReLU as the active function. Actually, for each $y_{l,i}\in y$, the convolution layer with kernel size one is calculated as: (4)$$y_{i,l}=\sum_{j=1}^{c}k_{l}x_{i,j}+b_{l},$$
where y is the output, $y_{i,l}$ is the *i*th *y* in the *l*th out-channel, x is the input with channel *c*, $x_{i,j}$ is the *i*th *x* in the *j*th in-channel, k represents the convolution kernel, $k_{l}$ is the *l*th channel of k and $b_{l}$ is the bias. This function can be seen as a FC layer operation in the channel dimension. Thus, in the transformation module, it is more convenient for the CNN to process the multi-channel data with fewer parameters and lower complexity than the FC layer.

### 2.2. Quantization

For a binary source, the BP algorithm is usually employed as the quantization for source compression. The principle of the BP quantization is based on the LDPC codebook satisfying $HC^{T}=0$ in GF(2), where C is the correct result, and the codebook H is the parity check matrix of the LDPC code.

However, the continuous source is quite different from the codebook of GF(2). If the continuous sequence is directly compressed by the BP based LDPC code, it will generate a larger distortion. Hence, a new quantizer based on the RMD strategy is designed to replace the BP. The RMD strategy is described in Algorithm 1. In this condition, the compression distortion is minimized to satisfy $HC^{T}=0$.

Firstly, the symbols in Algorithm 1 are defined as follows:

s: the input source data;

iter: the maximum number of iterations;

λ: the allocation of cost weight between variable and check nodes;

m: the mask vector, for which the masked nodes are set to zero;

q: the quantized s, and also the output of the RMD algorithm;

${\left(\cdot \right)}_{v},{\left(\cdot \right)}_{c}:$ the subscripts represent variable and check parts of symbol (·), respectively;

g(·): the generation function of the LDPC code;

**coe**, **map**: the coefficients of the RMD algorithm, and **map** contains **map**${}^{0}$ and **map**${}^{1}$;

**fc**: the cost vector of each node, and it contains **fc**${}^{s}$ and **fc**${}^{t}$;

V,C: the sets of variable and check nodes, respectively;

V(k): the variable nodes connected with the *k*-th check node;

C(k): the check nodes connected with the *k*-th variable node;

lr: the learning rate of the training stage;

[·|·]: merging of two variables;

argmin(·): the positioning function of the minimum element.

In addition, the map(·) function is calculated by
(5)$$\mathrm{map}\left(q\left[i\right]\right)=\left\{\begin{array}{cc} {\mathrm{map}}^{0}\left[i\right], & q\left[i\right]=0,\\ {\mathrm{map}}^{1}\left[i\right], & q\left[i\right]=1. \end{array}\right.$$
where ${\mathrm{map}}^{0}\left[i\right]$, ${\mathrm{map}}^{1}\left[i\right]$ and q[i] represent the *i*th element of ${\mathrm{map}}^{0}$, ${\mathrm{map}}^{1}$ and q, respectively.

In Algorithm 1, the variables q and map are initialized from lines 1 to 6. The variable **coe** is initialized according to **map** and the input mask m from lines 7 to 12. **fc**${}^{s}$ and ${\mathrm{fc}}^{t}$ are calculated from lines 13 to 17. In the while loop, $q_{c}\left[k\right]$ is flipped, and it is determined by the minimum ${\mathrm{fc}}_{c}^{t}\left[k\right]$. If ${\mathrm{fc}}_{c}^{t}\left[i\right]<0$, the flipping will reduce the distortion; then, $q_{v},{\mathrm{fc}}^{s}$ and ${\mathrm{fc}}^{t}$ need to be updated. From lines 29 to 31, **map** is updated by using the gradient descent with learning rate lr, and it is saved for the next use at line 33. When the RDM algorithm is not implemented at the training stage, lines 5 and 6 will be replaced by loading **map**, and lines 29 to 32 will be removed. Algorithm 2 presents the cost function of the RMD algorithm, which calculates the flip cost of each nodes and assigns them to **fc**${}^{s}$ according to **coe**. The flow chart of Algorithm 1 is shown in Figure 5.

**Algorithm 1** RMD algorithm.
**Input:**

H,s,iter,λ,m


**Output:**

q

  1: $s_{v},s_{c}\leftarrow s$  2: $q_{c}\leftarrow \mathrm{sign}\left(s_{c}\right)$  3: $q_{v}\leftarrow g\left(q_{c}\right)$  4: $q\leftarrow [q_{v}|q_{c}]$  5: **map**
←0  6: ${\mathrm{map}}^{1}\leftarrow {\mathrm{map}}^{1}+1$  7: **coe**
$\leftarrow 1/{\left({\mathrm{map}}^{0}-{\mathrm{map}}^{1}\right)}^{2}$  8: **for**
*i* in m **do**  9:    **if** m[i]=0 **then**10:         **coe**[i] ←011:     **end if**12: **end for**13: ${\mathrm{fc}}^{s}\leftarrow \cos t\left(q,s,\mathrm{coe}\right)$14: **fc**
${}_{v}^{s},$
**fc**
${}_{c}^{s}\leftarrow$
**fc**
${}^{s}$15: **for**
*i* in C **do**16:     **fc**${}_{c}^{t}\left[i\right]\leftarrow \lambda$**fc**${}_{c}^{s}\left[i\right]+\left(1-\lambda \right)\sum_{j\in V(i)}$**fc**${}_{v}^{s}\left[j\right]$17: **end for**18: n←019: **while** 
n<iter
**do**20:     n←n+121:     k←argmin(**fc**${}_{c}^{t})$22:     **if** ${\mathrm{fc}}_{c}^{t}\left[k\right]<0$ **then**23:         $q_{c}\left[k\right]\leftarrow 1-q_{c}\left[k\right]$24:         $q_{v},$**fc**${}^{s},$**fc**${}^{t}\leftarrow$ update$(s_{v},q_{v},k,$**fc**${}^{s},$**fc**${}^{t})$25:     **else**26:         **break**27:     **end if**28: **end while**29: $\hat{s}\leftarrow \mathrm{map}\left(q\right)$30: $\mathrm{gradient}\leftarrow 2(s-\hat{s})$31: map←map+gradient·lr32: save **map**33: **return** q

**Algorithm 2** ${\mathrm{fc}}^{s}=\cos t\left(q,s,\mathrm{coe}\right)$.
**Input: **

q,s,coe

**Output:** 
${\mathrm{fc}}^{s}$1: ${\mathrm{fc}}^{s}\leftarrow 0$2: **for** i in q **do**3:   ${\mathrm{fc}}^{s}\left[i\right]\leftarrow \mathrm{coe}\left[i\right]\cdot \left({\left(s-\mathrm{map}\left(1-q\left[i\right]\right)\right)}^{2}-{\left(s-\mathrm{map}\left(q\left[i\right]\right)\right)}^{2}\right)$4: **end for**5: **return** 
${\mathrm{fc}}^{s}$

In the RMD algorithm, *q* is flipped with the minimum ${\mathrm{fc}}^{t}$ in each iteration satisfying qH=0. Here, the minimum ${\mathrm{fc}}^{t}$ indicates the maximum quantization error between itself and the associated variable node; therefore, the flipping will effectively reduce the total quantization distortion. In the training process, **coe** and **map** are updated by gradient descent. With **coe** and **map** updating, the ${\mathrm{fc}}^{t}$ will be calculated more accurately.

Algorithm 3 presents a quick way to update $q_{v},{\mathrm{fc}}^{s}$ and ${\mathrm{fc}}^{t}$. Only if $q_{v}\left[i\right]$ satisfying i∈V(k) is flipped, the corresponding ${\mathrm{fc}}_{v}^{s}\left[i\right]$ can be updated. Then, the corresponding ${\mathrm{fc}}_{c}^{t}\left[j\right]$ is refreshed by calculating ${\mathrm{fc}}_{c}^{t}\left[j\right]={\mathrm{fc}}_{c}^{t}\left[j\right]+\left(1-\lambda \right)\left({\mathrm{fc}}_{v}^{s}\left[i\right]-t\right)$. In this case, it does not need to recalculate $q_{v},{\mathrm{fc}}^{s}$ and ${\mathrm{fc}}^{t}$. The flow chart of Algorithm 3 is shown in Figure 6.

**Algorithm 3** $q_{v},{\mathrm{fc}}^{s},{\mathrm{fc}}^{t}=\mathrm{update}\left(s_{v},q_{v},k,{\mathrm{fc}}^{s},{\mathrm{fc}}^{t}\right)$.
**Input: **

$s_{v},q_{v},k,{\mathrm{fc}}^{s},{\mathrm{fc}}^{t}$

 **Output:** $q_{v},{\mathrm{fc}}^{s},{\mathrm{fc}}^{t}$1: **for** i in V(k) **do**2:    $q_{v}\left[i\right]=1-q_{v}\left[i\right]$3:    $t\leftarrow {\mathrm{fc}}_{v}^{s}\left[i\right]$4:    ${\mathrm{fc}}_{v}^{s}\left[i\right]=\cos t\left(q_{v}\left[i\right],s_{v}\left[i\right]\right)$5:    **for** j in C(i) **do**6:       ${\mathrm{fc}}_{c}^{t}\left[j\right]={\mathrm{fc}}_{c}^{t}\left[j\right]+\left(1-\lambda \right)\left({\mathrm{fc}}_{v}^{s}\left[i\right]-t\right)$7:    **end for**8: **end for**9: **return** 
$q_{v},{\mathrm{fc}}^{s},{\mathrm{fc}}^{t}$

Each node *i* in the check matrix of the LDPC code with mask vector m[i]=1 is filled with $s_{m}^{\prime }\left[i\right]$ before the RMD training. The output q is compressed as $q_{c}$ following qH=0, and it can be reconstructed by $q=g(q_{c})$, where g(·) is the generation function of LDPC code. This allows the rate $r=\frac{n-k}{n-m^{\prime }}$ to be changed from (n−k)/n to 1 according to the variable $m^{\prime }$, where *n* and *k* are the code length and numbers of variable nodes, respectively, and $m^{\prime }$ represents the number of element 1 in mask vector m.

The computational complexity of the proposed RMD algorithm is
(6)$$\begin{array}{cc} O_{RMD} & =O_{initial}+O_{iterate}\\ \; & =O(n\times dv\times dc)+O(t\times dv\times dc)\\ \; & =O(n\times dv\times dc), \end{array}$$
where *n* is the number of check nodes; *t* is the number of iterations satisfying t<n; and dv and dc are the degrees of the variable and check nodes, respectively. In addition, the number of iterations is limited to 30 in the RMD algorithm, and the BP algorithm needs over 100 iterations. Overall, the computational and time complexities of the RMD algorithm are both lower than those of the BP algorithm.

### 2.3. Decoder

The decoder structure is shown in Figure 7. Referring to the encoding scheme, q is divided into $q_{1}∼q_{m}$, and they are input into the MLP. After that, $q_{1}^{\prime }∼q_{m}^{\prime }$ are unified as a matrix: (7)$$q_{1}^{\prime }⊕q_{2}^{\prime }⊕\ldots ⊕q_{m}^{\prime }\Rightarrow \left[\begin{array}{c} q_{1}^{\prime }\\ q_{2}^{\prime }\\ \vdots \\ q_{m}^{\prime } \end{array}\right].$$
Then, the joint result is sent to the CNN and reconstructs $\hat{s}$. The corresponding parameters and structure of the CNN can be referred to from Figure 4.

## 3. System Optimization and Technical Details

### 3.1. Gradient Backpropagation

In this section, the non-differentiability problems in the RMD algorithm and the neural network are resolved. Since the RMD algorithm is used as one layer of the neural networks, the backpropagation of this layer needs to provide the effective gradients. In this case, the coefficients of the neural networks are updated before the quantization according to the gradients, so that the loss function can be minimized.

However, in the quantization procedure, the values of the derivative function are mostly zeros. In this case, the gradient backpropagation of the neural network will be terminated [26]. To solve this problem, an existing work considers adding random noise to the quantization for training [14]. However, there is a larger discrepancy between the testing and training procedures, which significantly affects the quantization results.

According to [27], a gradient expectation is theoretically computed with the finite difference; i.e.,
(8)$$\begin{array}{cc} \frac{d}{ds}E\left[Q\left(s+u\right)\right]\; & =\frac{d}{ds}\int_{-t/2}^{t/2}Q\left(s+v\right)dv\\ \; & =Q(s+t/2)-Q(s-t/2), \end{array}$$
where Q(·) is the quantization function, E(·) is the expectation calculation, *t* is the length of granular cells of the quantizer and the distribution follows u∼U(−t/2,t/2). Equation (Equation 8) allows one to evaluate the derivative even if *Q* is non-differentiable. By extending the *Q* function to a vector s+u, where $u∼U{\left(-t/2,t/2\right)}^{D}$, and the superscript *D* represents the dimension of the input vector, it has
(9)$$\begin{array}{cc} \; & \frac{\partial }{\partial s_{i}}E\left[\left(Q\left(s+u\right)\right)\right]\\ \; & =E\left[\frac{\partial }{\partial z_{i}}{\left(Z\right)|}_{Z=Q(s+u)\cdot }\frac{\partial }{\partial s_{i}}Q\left(s_{i}+u_{i}\right)\right],\\ \; & \approx E\left[\frac{\partial }{\partial z_{i}}{\left(Z\right)|}_{Z=Q(s+u)}\right]\cdot E\left[\frac{\partial }{\partial s_{i}}Q\left(s_{i}+u_{i}\right)\right]. \end{array}$$
Here, Z is an independent variable at the next layer. From Equation (Equation 8), the derivative of the backpropagation is replaced by the following expectation:(10)$$\begin{array}{c} E\left[\frac{\partial }{\partial s_{i}}Q\left(s_{i}+u_{i}\right)\right]=Q\left(s_{i}+t/2\right)-Q\left(s_{i}-t/2\right). \end{array}$$

By replacing the original derivative function with the expectation value, the gradients from the next layer can correctly calculate the quantization output. In this way, the compression rate is converted in a larger interval before the quantization.

### 3.2. Training the Network

The proposed scheme can be seen as an end-to-end network, where the labels should be the input continuous sequences themselves, the mean square error (MSE) loss is selected as the loss function and Adam is the optimizer. In the network, the learning rate is set to 0.005, and the batch size is 1024; see Table 3 for details. However, if the MSE loss is only used on the output of the total system, the network is hard to converge. It is recommended to add the loss function to each $q_{i}^{\prime }$ after the MLP in the transformation module, which can speed up the network’s convergence.

## 4. Simulation Results and Discussions

In this section, we take Gaussian sources following the distribution N(0,1) as an example. By using the MSE measurement, the distortion-rate limit [28] is expressed as
(11)$$d=2^{-2r},$$
where *d* is the theoretical distortion, and *r* represents the compression rate in bits/symbol.

The check matrix of the P-LDPC code is extended by using the progressive edge-growth algorithm [29], and the compression rate *r* is calculated by
(12)$$\begin{array}{c} r=m-1+\frac{n-k}{n-m^{\prime }}, \end{array}$$
where m−1 indicates that there are m−1 quantizers, $Q_{1}∼Q_{m-1}$, which are set as the sign functions of rates 1, and the rate of $Q_{m}$ is $\frac{n-k}{n-m^{\prime }}$.

In Figure 8, the distortion-rate performances are analyzed based on different benchmark P-LDPC codes in [30], including AR3A, AR4JA and ARA codes. The extending times were 5 and 20. It can be seen that he code after extending five times had a better distortion-rate performance than the 20-times-extended code. That is, the fewer dimensions the check matrix uses, the more the system’s distortion is reduced. Note that the dimensions of the check matrix significantly increase the time consumption of the proposed scheme. Hence, the proposed system has lower complexity by employing less P-LDPC code.

In Figure 9, the distortion-rate performance is compared for the BP and the RMD algorithms. Three benchmark P-LDPC codes [30] were used for simulations. It is clear that the AR3A code achieved better results, approaching the distortion-rate limit. Furthermore, instead of the BP algorithm, the RMD algorithm using AR3A code was closer to the distortion-rate limit. Hence, the RMD algorithm is more efficient than the BP algorithm.

In Figure 10, the distortion-rate performance is shown at the high-rate regime. When the original derivative function is replaced by the new one, the neural network obtains correct gradients to update the coefficients. In this case, it is obvious that the simulation with the new derivative function is closer to the distortion-rate limit. In the rate interval from 0 to 1, these two curves approach one another, since the feedback mechanism does not need to work. Overall, the replaced derivative function and feedback mechanism ensure the system to work well when the rate goes higher.

In Figure 11, the proposed scheme, is further compared with the MLC system [13]. By using the AR3A code, it is clear that the proposed system brings a performance improvement over the MLC scheme. Even though an optimally-designed code in [13] is implemented by the MLC system, its performance is still worse than the NN-LP-LDPC. Therefore, the NN-LP-LDPC code is not only an efficient system for the lossy compression, but it also has simpler structure than the MLC system.

In addition, since the proposed scheme was designed based on neural networks, it is demonstrated that the system input is applicable to a general source—for example, the continuous sequences following Gaussian, Laplacian and other distributions. The related simulations are shown in Figure 12. The Laplacian source follows $f\left(x\right)=\frac{\lambda }{2}e^{-\lambda |x|}$, and the distortion-rate limit of the Laplacian source with the MSE distortion is [31]: (13)$$D_{\delta }=\frac{2}{\lambda^{2}}-\frac{\Delta }{\lambda }\left(1+\coth\frac{\lambda \Delta }{2}\right)e^{-\frac{\lambda \Delta }{2}},$$
(14)$$R_{\delta }=-p^{\prime }\left(0\right)\log_{2}p^{\prime }\left(0\right)-e^{-\frac{\lambda \Delta }{2}}\log_{2}\sinh\frac{\lambda \Delta }{2}+\frac{\lambda }{\log2}S,$$
where the distortion-rate limit is expressed as a parametric equation, the parametric is Δ, Δ∈(0,+∞), $p^{\prime }\left(0\right)=1-e^{-\frac{\lambda \Delta }{2}}$, and *S* is calculated by
(15)$$S=2\Delta \cdot \sinh\frac{\lambda \Delta }{2}\sum_{i=0}^{n}ie^{-i\lambda \Delta }.$$
In our system simulation, the variance of the Laplacian source is given as $\frac{2}{\lambda^{2}}=1$, and λ is set to $\sqrt{2}$.

It is clear that both the Gaussian and the Laplace sources have good performances to approach the distortion-rate limit. Furthermore, the simulating performances of the two types of sources were similar, which indicates that the proposed scheme has good robustness for different source distributions.

## 5. Conclusions

In this paper, it is demonstrated that the new route of “transform-quantization” significantly outperforms the conventional “quantization-compression” by using the neural networks. This provides a different method with which to design the lossy compression system. In addition, the P-LDPC codes were inserted into the neural networks as the NN-LP-LDPC system, which is obviously different from the existing work. The effectiveness of the proposed scheme was verified by simulation results. Compared with the existing works, the proposed scheme achieved both better performance and lower complexity. Furthermore, it has versatility and is suitable for compressing different sources. However, due to its simple structure, one drawback is that the current scheme may not work well for image/video compression, which is left as future work. In addition, the P-LDPC codes used in this paper are not optimized for lossy compression. Our future work will focus on the system optimizations, including the design of P-LDPC codebooks, the improvement of the RMD algorithm and the design of practical neural networks. 

## Figures and Tables

**Figure 1 entropy-25-00252-f001:**
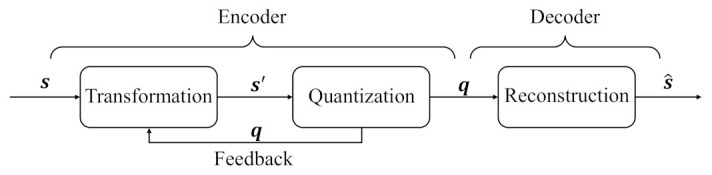
The “transformation-quantization” route based on the NN-LP-LDPC system.

**Figure 2 entropy-25-00252-f002:**
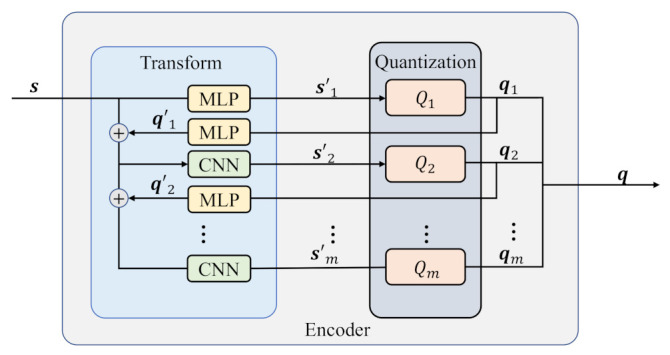
The encoder of proposed scheme: the transformation module contains MLP and CNN networks, and the quantization module consists of multiple quantizers.

**Figure 3 entropy-25-00252-f003:**
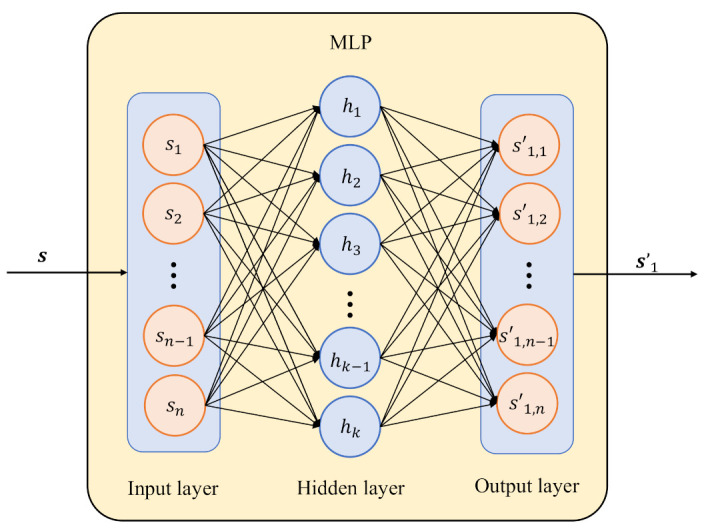
The MLP with the structure of the FC layer.

**Figure 4 entropy-25-00252-f004:**
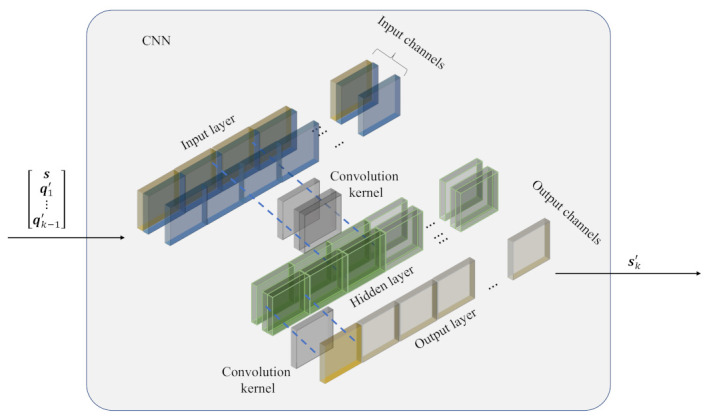
The CNN with 1D convolution layer, and its kernel size is 1.

**Figure 5 entropy-25-00252-f005:**
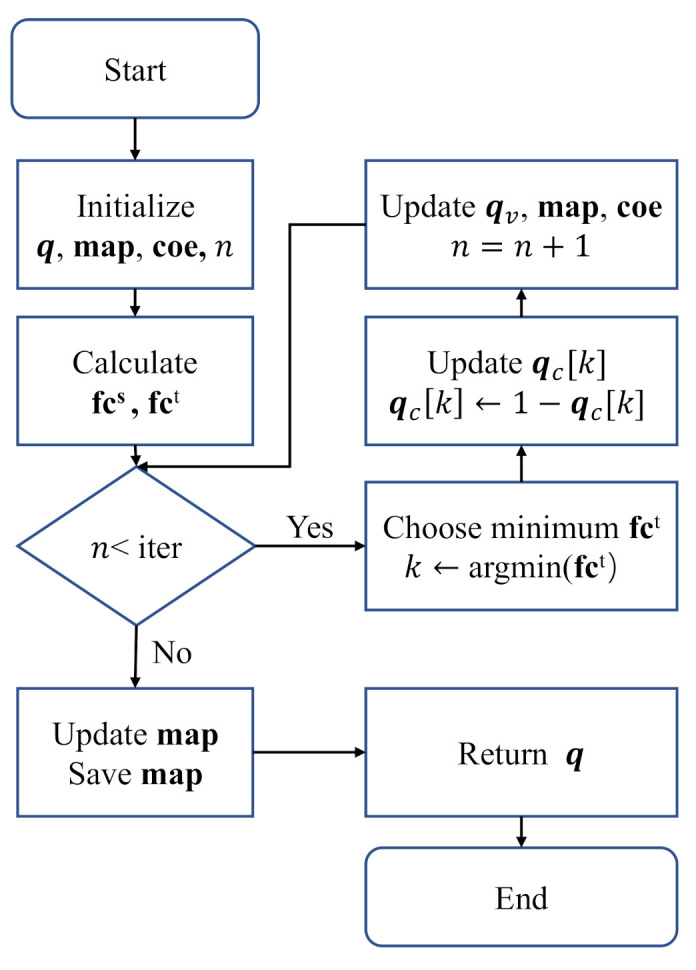
The flow chart of Algorithm 1.

**Figure 6 entropy-25-00252-f006:**
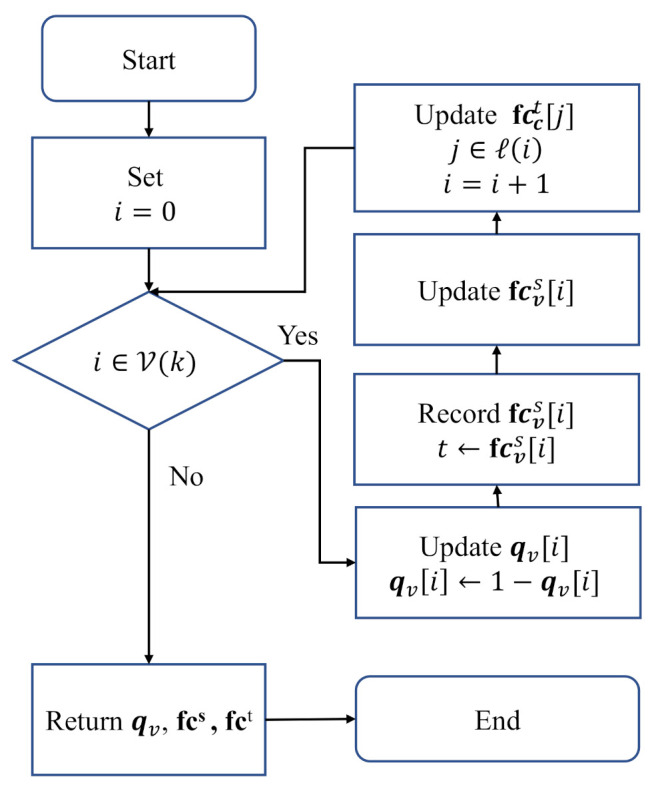
The flow chart of Algorithm 3.

**Figure 7 entropy-25-00252-f007:**
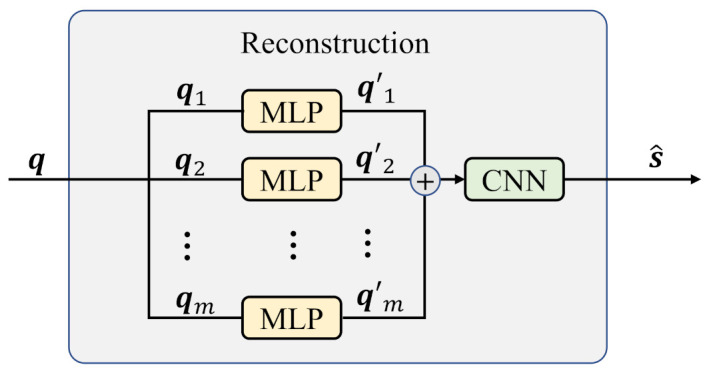
The decoder of the proposed scheme.

**Figure 8 entropy-25-00252-f008:**
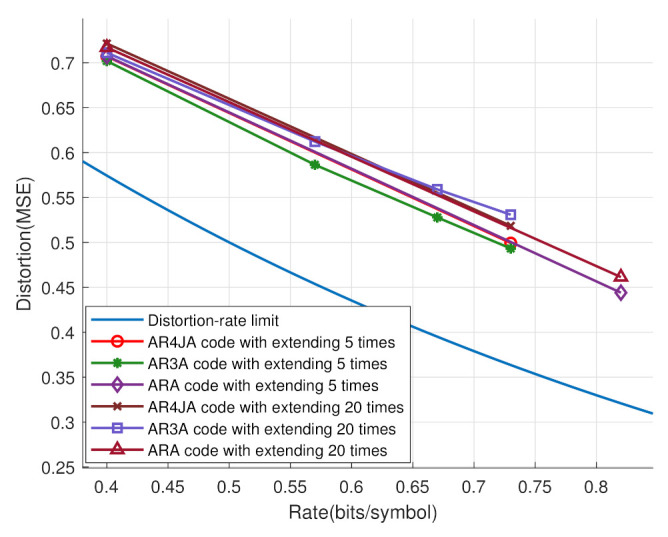
The distortion-rate analyses based on different P-LDPC codes and extending times.

**Figure 9 entropy-25-00252-f009:**
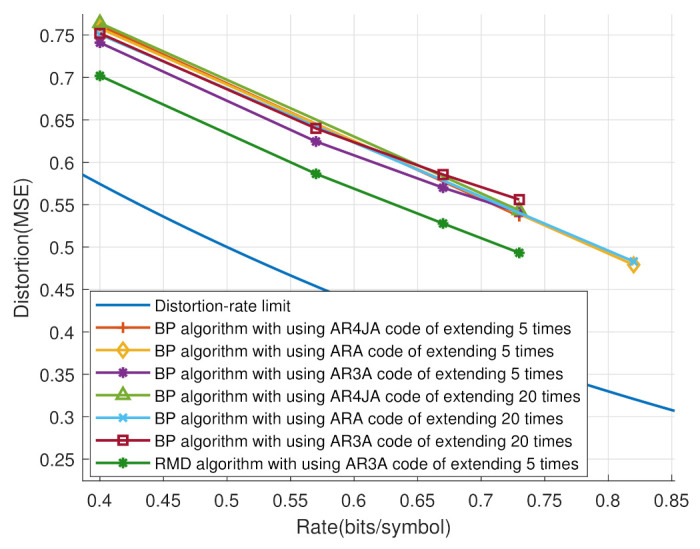
The distortion-rate comparison between the BP and RMD algorithms.

**Figure 10 entropy-25-00252-f010:**
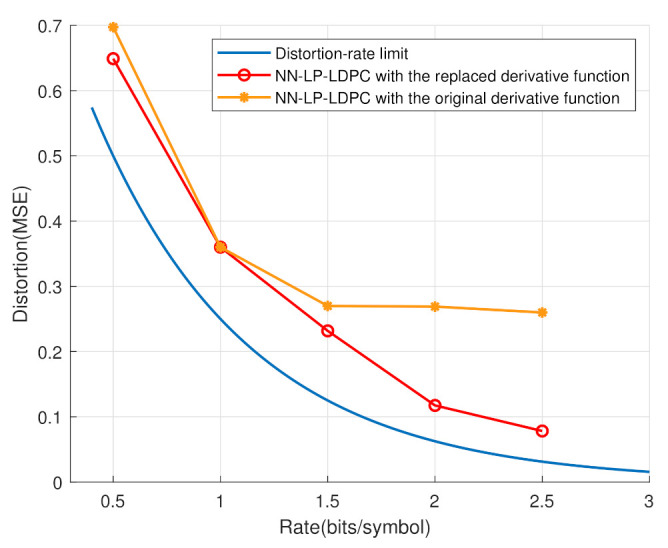
The distortion-rate analyses of the proposed scheme based on the distinct derivative functions.

**Figure 11 entropy-25-00252-f011:**
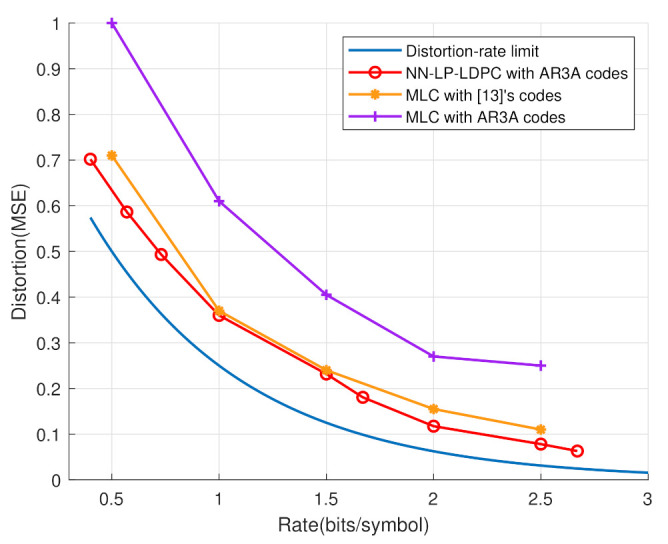
The distortion-rate comparison between the NN-LP-LDPC and MLC [13] systems.

**Figure 12 entropy-25-00252-f012:**
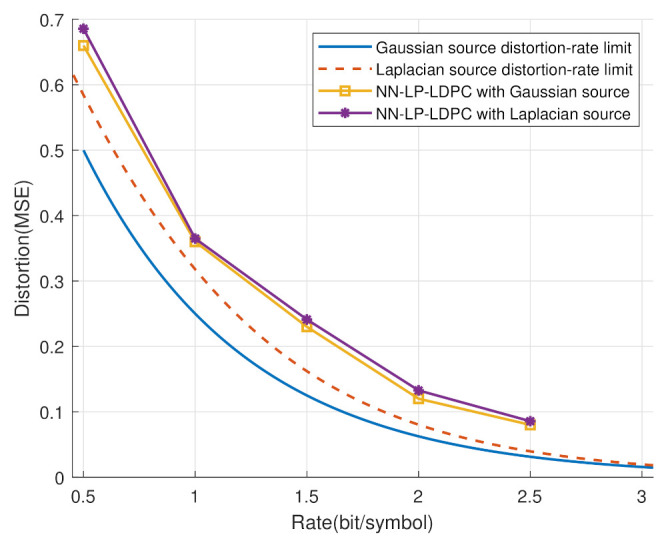
The distortion-rate analyses of the general source, including the Gaussian and the Laplacian sources.

**Table 1 entropy-25-00252-t001:** Literature description.

Literature	Main Contribution
Braunstein [9]	Lossy compression of binary sources using reinforced belief propagation decoding algorithm of LDPC
Fang [10]	Lossy compression of binary source using sliding-window BP decoding algorithm of LDPC
Liu [11]	Use P-LDPC code for binary source compression
Wang [12]	Performance of binary source lossy compression using P-LDPC in AWGN channel
Deng [13]	Use P-LDPC code for Gaussian source compression
Proposed scheme	Designed the RMD algorithm, combining the neural network with P-LDPC, and realized the lossy compression of general information sources

**Table 2 entropy-25-00252-t002:** Literature comparison.

Literature	LDPC Type	Method	Sources
Braunstein [9]	LDPC	RBP	Binary source
Fang [10]	LDPC	sliding-window BP	Binary source
Liu [11], Wang [12]	P-LDPC	RBP	Bianry source
Deng [13]	P-LDPC	MLC and RBP	Gaussian source
Proposed scheme	P-LDPC	Tranformation and RMD	General source

**Table 3 entropy-25-00252-t003:** Training parameters.

Loss	Learning Rate	Optimizer	Batch Size	Number of Epochs
MSE	$5\times 10^{-3}$	Adam	1024	1000

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
