# Peer review of "Lossy P-LDPC Codes for Compressing General Sources Using Neural Networks"

_entropy, 2023, doi:10.3390/e25020252_

Round 1

Reviewer 1 Report

This paper studies a recently hot topic, i.e., lossy source coding with artificial neural network. A novel framework was proposed, accompanied with technical details. This paper is interesting and technically sound. I recommend to accept it in present form.

A minor comment: Currently the activator at each neuron is the ReLU function. As we know, the sigmoid function does not work for deep neural work, so it is replaced by the ReLU function. Fig. 3 shows a shallow network. I wonder whether the sigmoid function works for this network?

Reviewer 2 Report

This paper talks about a new "transform-quantization" compression system instead of the traditional "quantization-compression" scheme.

Here are some concerns.

1. Maybe the authors can add a related work session to discuss related work, e.g., the recent topic of 'neural compression' and neural network-based data compression methods.

2. It is not very clear why the authors choose the MLP+CNN neural network structure for the transformation task. Maybe some insights can be given into this. How about other neural network structures, such as RNN, attention network, etc.? 

3. In the experiment, only a Gaussian source is considered. How about other sources, e.g., binary sources, images, videos, etc.? Maybe some more results can be shown.

Reviewer 3 Report

 In this paper authors have proposed a lossy compression scheme for general continuous sources. My Review is as under.

1.    The abstract should be revised, and the recommended technique should be stated briefly. Kindly update the paragraph “lossy compression scheme is proposed for general continuous sources, e.g., Gaussian and Laplacian sources”.

2.    Include one generic diagram in the Introduction Part to help readers comprehend.

3.    Include one table relating to the literature analyses.

4.    “NN-LP-LDPC system” may be replaced by “proposed model/scheme”.

5.    Hidden layer in Figure 3. Must be redrawn as Figure 4.

6.    Kindly add flow charts for Algorithm 1, 2 and 3.

7.    Simulation and analysis parameters must be referenced.

8.    The conclusion should be rewritten, and please include the drawbacks and advantages of the suggested approach.

9.    Kindly include more references from years 2020-2022
